# Estimating Yield from NDVI, Weather Data, and Soil Water Depletion for Sugar Beet and Potato in Northern Belgium

Astrid Vannoppen [1,*] and Anne Gobin [1,2]

1    Vlaamse Instelling Voor Technologisch Onderzoek NV, 2400 Mol, Belgium; anne.gobin@vito.be
2    Department of Earth and Environmental Sciences, Faculty of Bioscience Engineering, University of Leuven, 3001 Leuven, Belgium
*    Correspondence: astrid.vannoppen@vito.be

**Abstract:** Crop-yield models based on vegetation indices such as the normalized difference vegetation index (NDVI) have been developed to monitor crop yield at higher spatial and temporal resolutions compared to agricultural statistical data. We evaluated the model performance of NDVI-based random forest models for sugar beet and potato farm yields in northern Belgium during 2016–2018. We also evaluated whether weather variables and root-zone soil water depletion during the growing season improved the model performance. The NDVI integral did not explain early and late potato yield variability and only partly explained sugar-beet yield variability. The NDVI series of early and late potato crops were not sensitive enough to yield affecting weather and soil water conditions. We found that water-saturated conditions early in the growing season and elevated temperatures late in the growing season explained a large part of the sugar-beet and late-potato yield variability. The NDVI integral in combination with monthly precipitation, maximum temperature, and root-zone soil water depletion during the growing season explained farm-scale sugar beet ($R^2 = 0.84$, MSE = 48.8) and late potato ($R^2 = 0.56$, MSE = 57.3) yield variability well from 2016 to 2018 in northern Belgium.

**Keywords:** root-zone soil water depletion; AquaCrop-OSPy; sugar beet; potato; crop yield; NDVI; Belgium; weather impact; random forest

## 1. Introduction

Information on how crop yield varies from year to year at field and global level is important for planning purposes. Farmers use crop-yield information to detect yield anomalies caused by varying environmental conditions during the crop growing season and to evaluate the effect of management choices on crop yield. For policy and decisionmakers, crop-yield data are imperative to make informed and strategic decisions on food and feed stocks [1,2]. Agricultural insurers need crop-yield information to get insights in the risk of negative impact of (extreme) weather on cropping systems and yield anomalies [3]. For these purposes, crop-yield data are ideally available at high spatial and temporal resolutions, which is not the case for agricultural yield statistics as they are typically available on a yearly basis and at the regional or country scale [4]. These crop-yield statistics provide coarse-scale information on local crop yields and may not be suitable to establish differences caused by local environmental conditions [5,6].

Crop-yield models have been developed to close the data gap between field and regional-scale crop yields. Crop-yield models simulate how crops grow in interaction with their environment. Remotely sensed vegetation indices can be included in empirical crop-yield models using statistical methods such as linear regression or random forests [7]. Empirical crop models based on vegetation indices provide crop-yield data at a high spatial resolution with coefficients of determination ($R^2$) ranging from 0.14 to 0.88 [8–14]. A vegetation index that has resulted in reasonable crop-yield estimations is the normalized difference vegetation index (NDVI) as an indicator of photosynthetic active biomass [4,8,9,15–19].

Time-series NDVI at different crop-growth stages or throughout the growing season has proven a good predictor in crop-yield models.

Crop-yield models that include weather information, in addition to a remotely sensed vegetation index, achieved higher model performances and explained up to 66% and 97% of yield variability at field and regional level, respectively [8–10]. Information on the soil water status throughout the growing season could potentially narrow the range of crop-yield model performance at the field level. An excess or scarcity of soil water, certainly during important phenological crop-growth stages, has a high impact on crop-yield quantity and quality [20,21]. Extreme weather events and long-term climate effects, which will impact the soil water available to crops, will have a large impact on agricultural yield in the future [22–24]. Therefore, including weather and soil water data in crop models will become important in a changing climate.

For tuber crops such as potato and sugar-beet vegetation index-based crop-yield models have been developed [25,26]. Vegetation indices such as the NDVI provide information on the functioning of the source leaves, which are pivotal in capturing the light and $CO_2$ needed for the growth of sink organs (taproot yield) and determine the crop yield of tuber crops like potato and sugar beet. The performance of empirical potato and sugar-beet crop-yield models based on the remotely sensed vegetation index NDVI has not yet been evaluated in northern Belgium. However, in previous research we found that model evaluation metrics of winter-wheat yield models based on NDVI yield proxies are dependent on the location. For northern Belgium NDVI-based winter-wheat yield models had a poor model prediction compared to Latvia, suggesting that NDVI does not capture winter-wheat yield variability well in northern Belgium [8,9]. We concluded that modeling winter-wheat yield based on NDVI using an empirical model is dependent on the crop's environment.

In this research we evaluate yield estimation based on the remotely sensed vegetation index NDVI and weather data for the crops sugar beet, late potato, and early potato in northern Belgium. In addition, we hypothesized that information on root-zone soil water depletion throughout the growing season improved yield modeling of sugar beet and late potato. Root-zone soil water depletion was modeled throughout the crop growing season based on crop specific parameters, soil texture, and weather data (i.e., minimum and maximum temperature, precipitation, and reference evapotranspiration) for each field using AquaCrop-OSPy [27,28].

## 2. Materials and Methods

### 2.1. Study Area

The locations, parcel information, and reported yields at farm level of sugar beet, late potato, and early potato for the years 2016, 2017, and 2018 were available from the Ministry of Agriculture. Only fields with an area bigger than 900 m² were considered to make sure that the extracted NDVI series were based on pure pixels. In total 468 sugar-beet fields, 685 late-potato fields, and 38 early-potato fields were used in the analysis (Figure 1).

### 2.2. Remote Sensing: NDVI Data

For each field, Sentinel-2-derived NDVI timeseries were extracted using the https://openeo.org/platform, accessed on 1 July 2021. The platform was used to apply a cloud mask to the five daily images based on the scene classification layer from Sentinel-2 and to calculate the average NDVI series based on the pixels that lie within the 10 m buffered fields. By applying this procedure, cloud-free timeseries were extracted for each field.

The NDVI integral (aNDVI) was calculated for each field using the trapezoidal rule [8]. NDVI values between day of year (DOY) 91–273 (i.e., the beginning of April to the end of September) for sugar beet, 121–273 (i.e., end of April to end of September) for late potato, and 91–212 (i.e., beginning of April to end of July) for early potato were considered for the calculation of aNDVI. NDVI values below 0.2 were discarded following [8,29]. In addition, fields with fewer than five NDVI observations and gaps of more than 60 days were not considered for the calculation of aNDVI. The number of sugar-beet, late-potato,

and early-potato fields for which aNDVI was calculated in 2016, 2017, and 2018 is presented in Table 1.

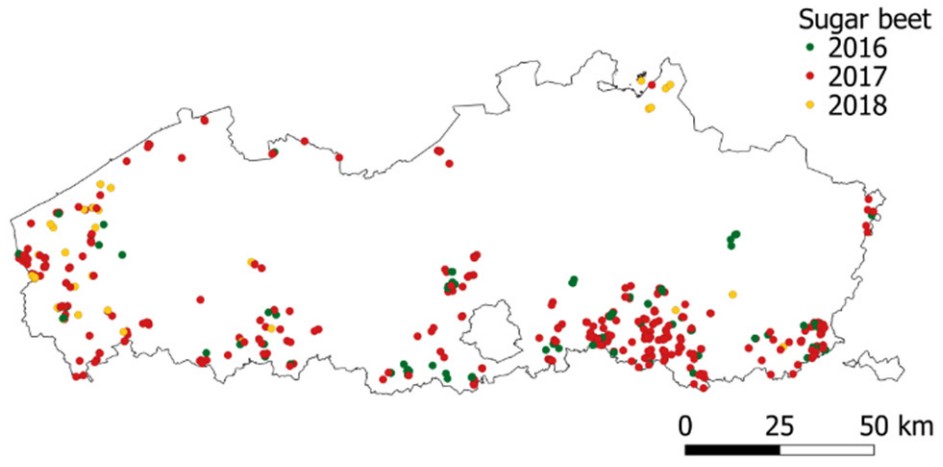

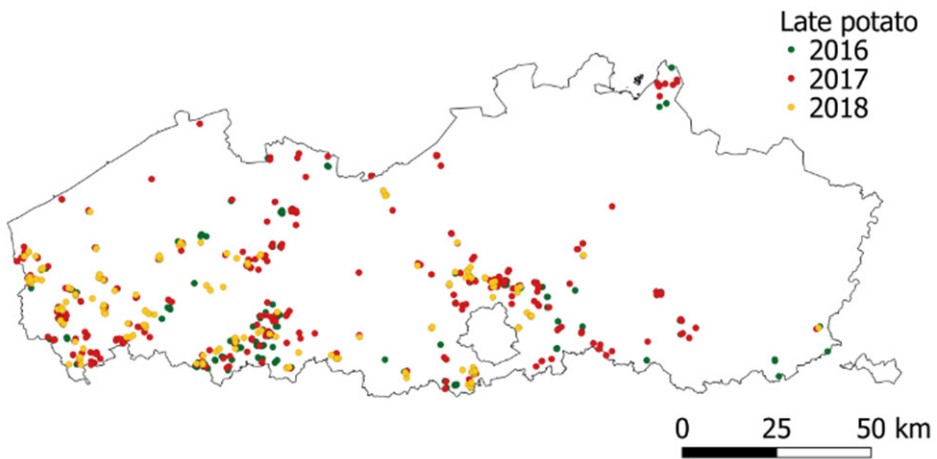

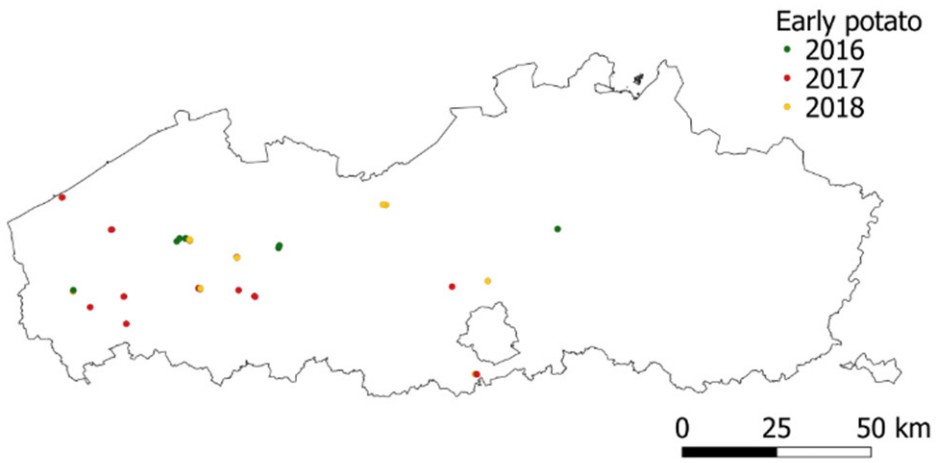

**Figure 1.** Locations of sugar beet (468 fields), late potato (685 fields), and early potato (38 fields) for the years 2016, 2017, and 2018 in northern Belgium.

**Table 1.** Sugar-beet, late-potato, and early-potato fields for which aNDVI was calculated. The numbers in parentheses are the number of farms to which the fields belong.

|  | Sugar Beet | Late Potato | Early Potato |
|---|---|---|---|
| 2016 | 92 (45 farms) | 149 (62 farms) | 8 (4 farms) |
| 2017 | 335 (122 farms) | 358 (112 farms) | 19 (11 farms) |
| 2018 | 41 (23 farms) | 178 (51 farms) | 11 (8 farms) |
| Total | 468 | 685 | 38 |

*2.3. Root-Zone Soil Water Depletion*

For sugar-beet and late-potato fields, the daily root-zone soil water depletion (i.e., SDrz) was calculated for each field using AquaCrop-OSPy [28]. AquaCrop-OSPy was used to model the soil water balance of the root zone for each field. AquaCrop-OSPy uses information on the soil texture, minimum and maximum temperature, reference evapotranspiration, and crop-specific parameters calibrated for northern Belgium to model the soil water balance in the root zone for each field [30]. Since calibrated crop-specific parameters were not available for early potato in northern Belgium, SDrz was not considered for this crop. Soil texture data were extracted for each field from the World Reference Base soil map of northern Belgium [31]. Information on the soil texture was used to derive the volumetric water content at field capacity, permanent wilting point, and saturation, and the saturated hydraulic conductivity (Ksat) of the soil root zone using pedo-transfer functions [32]. These variables were used to derive parameters governing soil evaporation, internal drainage and deep percolation, surface runoff, and capillary rise in AquaCrop-OSPy [33]. Weather data were extracted for each field from a 5 km resolution grid provided by the Royal Meteorological Institute [34]. The root zone was considered as a reservoir with incoming water fluxes from rainfall, and outgoing water fluxes from runoff, evapotranspiration, and deep percolation [30]. The water retained in the root zone, and the root-zone soil water depletion (i.e., SDrz) throughout the growing season were modeled with the soil water balance using AquaCrop-OSPy [30,35]. Irrigation was assumed to be zero for all fields. Sugar beet and potato are not often irrigated in northern Belgium—only under extremely dry circumstances farmers might apply supplementary irrigation. This is because the financial gain in yields when irrigated does not compensate for the cost of irrigation [36]. The SDrz refers to the amount of water that is required to bring the water amount in the root zone back to field capacity [30,33]. The field capacity expresses the maximum amount of water that can be retained against gravitational forces [30,33]. Thus, higher (lower) values of SDrz indicate that more (less) water is required to bring the amount of water in the root zone to field capacity. Negative SDrz values indicate that the soil water content in the root zone exceeds field capacity, thus approaching soil saturation. The daily SDrz were summed for each month during the growing season and were used as predictor variables in the sugar-beet and late-potato crop models (see Section 2.4.1).

*2.4. Data Analysis*

2.4.1. Yield Model

Yield models were built using a random-forest approach based on 500 trees. In a first model, aNDVI was the only predictor variable to model crop yield. In a second model, the weather variables monthly precipitation (P) and maximum temperature (Tmax) during the growing season were added to the yield model. The out-of-bag prediction error (MSE) and the explained variance ($R^2$) computed on the out-of-bag data were used to evaluate the model performance of the random forests. The R package ranger was used to build the random forests regression models.

For sugar beet and late potato, a third model was built, where yield was simulated in function of aNDVI and monthly SDrz during the growing season. These models allowed us to evaluate whether adding soil-water information improved the crop-yield model for sugar beet and late potato. Finally, a fourth model was built for sugar beet and late potato, where

crop yield was modeled in the functions of aNDVI and the monthly P, Tmax, and SDrz during the growing season. The importance of the predictor variables was calculated for this last model to determine which predictors explained most of the crop-yield variability.

### 2.4.2. Effect of Environmental Variables and Crop Yield

The Pearson correlation between monthly environmental variables (i.e., SDrz, P, and Tmax) and yield data was calculated for the growing season of sugar beet, late potato, and early potato fields. A significance level of 0.01 was used to identify environmental variables that are correlated with sugar beet, late potato, and early potato yields. In addition, the Pearson correlation between the monthly environmental variables and aNDVI was calculated. The correlation patterns between environmental variables and yield, and environmental variables and aNDVI were compared to determine to what extents crop yield and aNDVI are influenced by the considered environmental variables.

## 3. Results

### 3.1. NDVI Series of Sugar Beet, Late Potato, and Early Potato

The average NDVI series for sugar beet, late potato, and early potato between 2016 and 2018 in northern Belgium are presented in Figure 2. The horizontal lines represent the start and end in the day of year (DOY) selected for the calculation of the aNDVI. The intervals 91–273 (i.e., the beginning of April to the end of September) for sugar beet, 121–273 for late potato (i.e., the end of April to the end of September), and 91–212 (i.e., the beginning of April to the end July) for early potato were considered for the calculation of aNDVI.

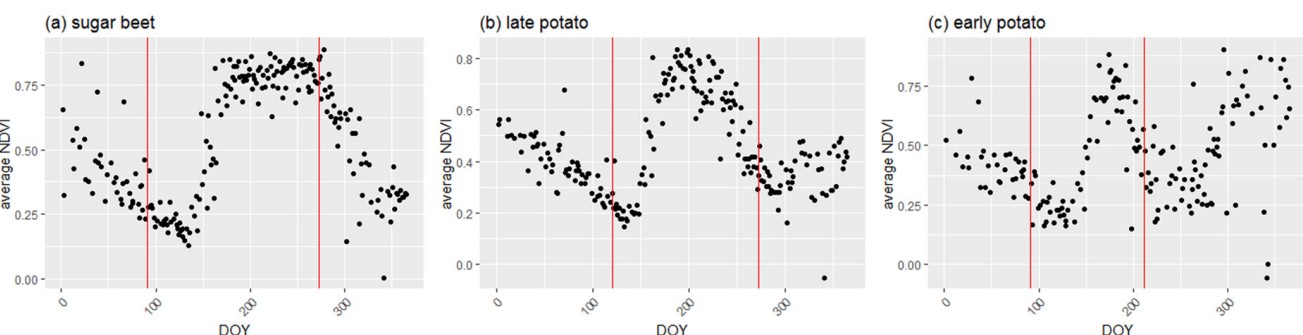

**Figure 2.** Average NDVI for (**a**) sugar-beet fields (468 fields), (**b**) late-potato fields (685 fields) and (**c**) early-potato fields (38 fields) from 2016–2018. The vertical red lines indicate the start and end DOY (day of year) used for the calculation of aNDVI: 91–273 for sugar beet (from the beginning of April to the end of September), 121–273 for late potato (from the end of April to the end of September), and 91–212 for early potato (from the beginning of April to the end of July).

### 3.2. aNDVI and Yield of Sugar Beet, Late Potato, and Early Potato

The boxplots of yield and the calculated aNDVI during 2016–2018 show that the variability in yield and aNDVI of the early-potato fields do not follow the same pattern (Figure 3c, right versus left graph). This is likely due to the low number of fields and farms in the sample. For sugar beet and late potato, the yield variability and aNDVI follow more similar patterns during 2016–2018 (Figure 3a,b, right versus left graph).

### 3.3. Random Forest Models and Variable Importance

The model performance indicators for the random forest yield models based on aNDVI indicate that aNDVI does not explain yield variability well for late and early potato (Model 1, Table 2). For sugar beet, the model performance of the aNDVI-based model was higher. When monthly weather data were added to the random forest yield models, the model performances increased for all crops (Model 2, Table 2). This indicates that weather variables were important yield predictor variables for sugar beet, late potato, and early potato.

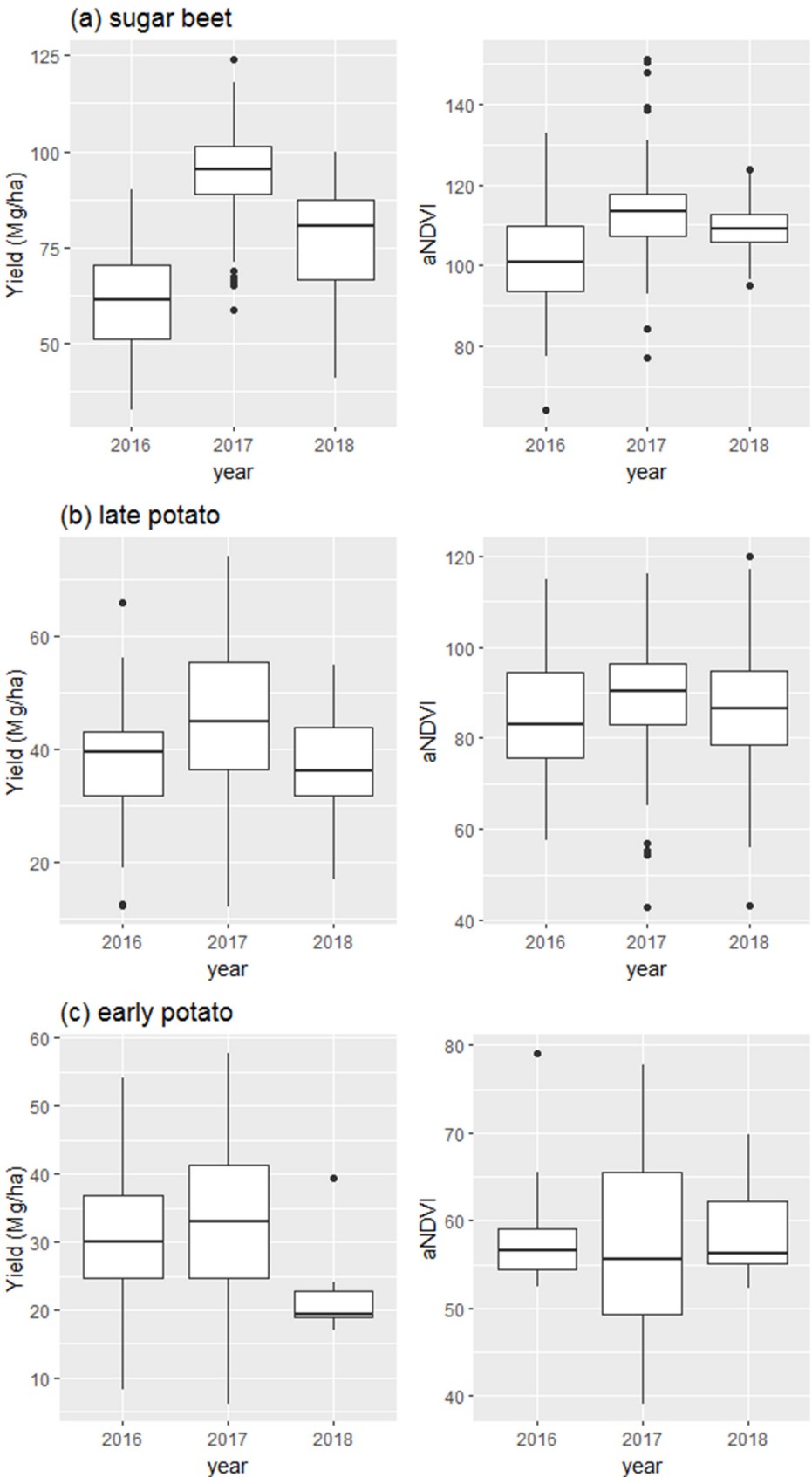

**Figure 3.** Boxplots of yield (Mg/ha, left graph) and aNDVI (right graph) for (**a**) sugar-beet fields (468 fields), (**b**) late-potato fields (685 fields), and (**c**) early-potato fields (38 fields) during 2016–2018.

**Table 2.** Model performance indicators for the random forest models based on monthly environmental variables. P: precipitation, Tmax: maximum temperature, SDrz: root-zone soil water depletion.

| Model 1: Yield—aNDVI | | | |
|---|---|---|---|
| | **Sugar Beet** | **Late Potato** | **Early Potato** |
| $R^2$ (out of bag) | 0.16 | −0.15 | 0.07 |
| MSE (out of bag) | 261.8 | 153.1 | 162.2 |
| Model 2: Yield—aNDVI + Monthly P + Monthly Tmax | | | |
| | **Sugar Beet** | **Late Potato** | **Early Potato** |
| Months in which P and Tmax were included in the random forest model | April–September | May–September | April–July |
| $R^2$ (out of bag) | 0.85 | 0.57 | 0.68 |
| MSE (out of bag) | 46.6 | 55.7 | 55.5 |
| Model 3: Yield—aNDVI + Monthly SDrz | | | |
| | **Sugar Beet** | **Late Potato** | |
| Months in which SDrz was included in the random forest model | April–September | May–September | |
| $R^2$ (out of bag) | 0.83 | 0.53 | |
| MSE (out of bag) | 54.4 | 61.9 | |
| Model 4: Yield—aNDVI + Monthly P + Monthly Tmax + Monthly SDrz | | | |
| | **Sugar Beet** | **Late Potato** | |
| Months in which P, Tmax and SDrz were included in the random forest model | April–September | May–September | |
| $R^2$ (out of bag) | 0.84 | 0.56 | |
| MSE (out of bag) | 48.8 | 57.3 | |

Information on root-zone soil water depletion throughout the growing season of sugar beet and late potato in combination with aNDVI explains sugar-beet and late-potato variability well (Model 3, Table 2). Model performances were similar to when sugar-beet and late-potato yields were modeled using weather variables throughout the growing season and aNDVI (Models 2 and 3, Table 2). When both weather variables and root-zone soil water depletion throughout the growing season were added to the sugar-beet and late-potato yield models, similar performances were reached compared to when only weather variables or root-zone soil water depletion in combination with aNDVI were used as predictors in the yield models (Model 4 versus Models 2 and 3, Table 2). The variable importance plot of the sugar-beet model based on aNDVI, weather variables, and root-zone soil water depletion throughout the growing season indicated that the root-zone soil water depletion in the month of April explained a large part of the sugar-beet yield variability (Figure 4a). In addition, aNDVI and maximum temperature in September were important variables (Figure 4a). Maximum temperature in September, root-zone soil water depletion in June and aNDVI were the most important variables to explain late potato yield variability (Figure 4b). The modeled versus predicted yields of the random forest models with predictor variables aNDVI, weather variables and root-zone soil water depletion throughout the growing season for sugar beet and late potato indicated that the models were able to predict sugar-beet and potato yield well (Figure 5). In Figure 6 the scatterplots of the environmental variables that explained a large part of the sugar-beet and late-potato yield versus crop yield in these models are shown.

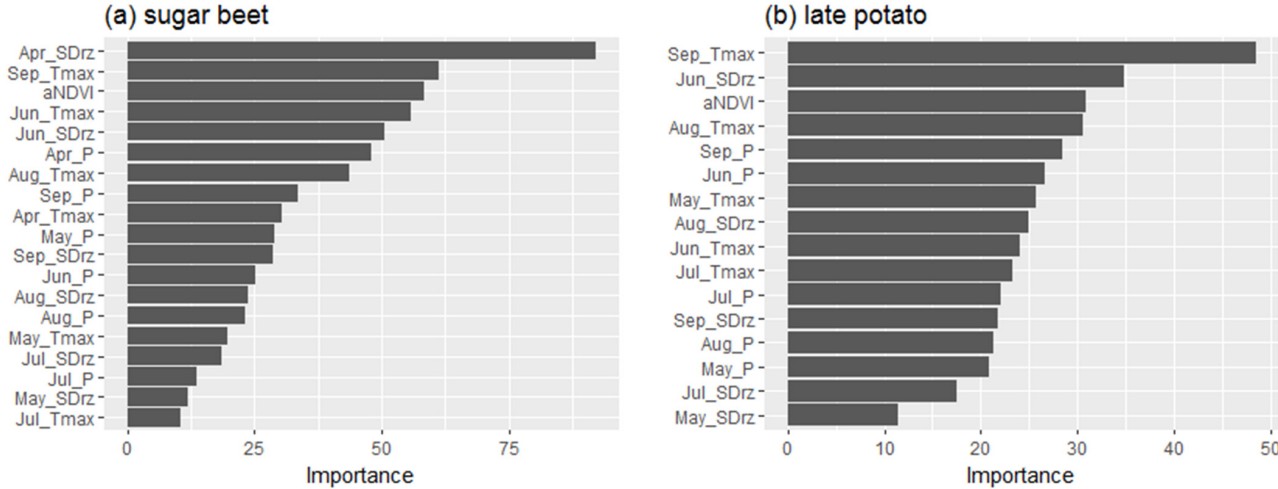

**Figure 4.** (**a**) Variable importance plot of Model 4 for sugar beet, (**b**) variable importance plot of Model 4 for late potato. P: precipitation, Tmax: maximum temperature, SDrz: root-zone soil water depletion.

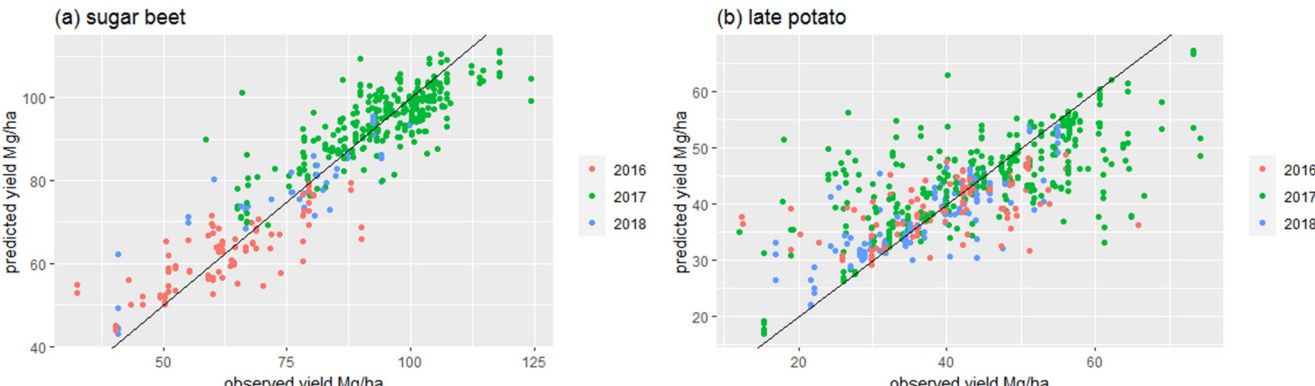

**Figure 5.** Modeled versus predicted yield for (**a**) sugar beet and (**b**) late potato as predicted by the random forest Model 4: yield = approximately aNDVI + monthly P + monthly Tmax + monthly SDrz (see Table 2 for model performance indicators and further details).

*3.4. Pearson Correlation Plots*

The Pearson correlation between the environmental variables of monthly precipitation, maximum monthly temperature, monthly root-zone soil water depletion, and crop yield for sugar beet, late potato, and early potato are presented in Figures 7a, 8a and 9a. The Pearson correlation between the environmental variables of monthly precipitation, monthly maximum temperature, monthly root-zone soil water depletion and the calculated aNDVI for sugar beet, late potato, and early potato are presented in Figures 7b, 8b and 9b. The correlation pattern between yield and the monthly environmental variables, and aNDVI and the monthly weather variables are similar for sugar beet, late potato, and early potato. However, the correlation values between monthly environmental variables and crop yield were higher compared to the correlation values between monthly environmental variables and the aNDVI for all crops (Figures 7–9). Correlation values between early-potato yield and monthly environmental variables were only significant for Tmax in April, May, and July. Correlation values between early-potato aNDVI and monthly environmental variables were not significant. The significant negative correlation between Tmax in September and yields of sugar beet and late potato was likely due to the effect of cumulative drought stress during the months of August and September. Fields that experienced a high Tmax in September also had a high Tmax in August, which in turn was also negatively correlated with yield (Figure 10).

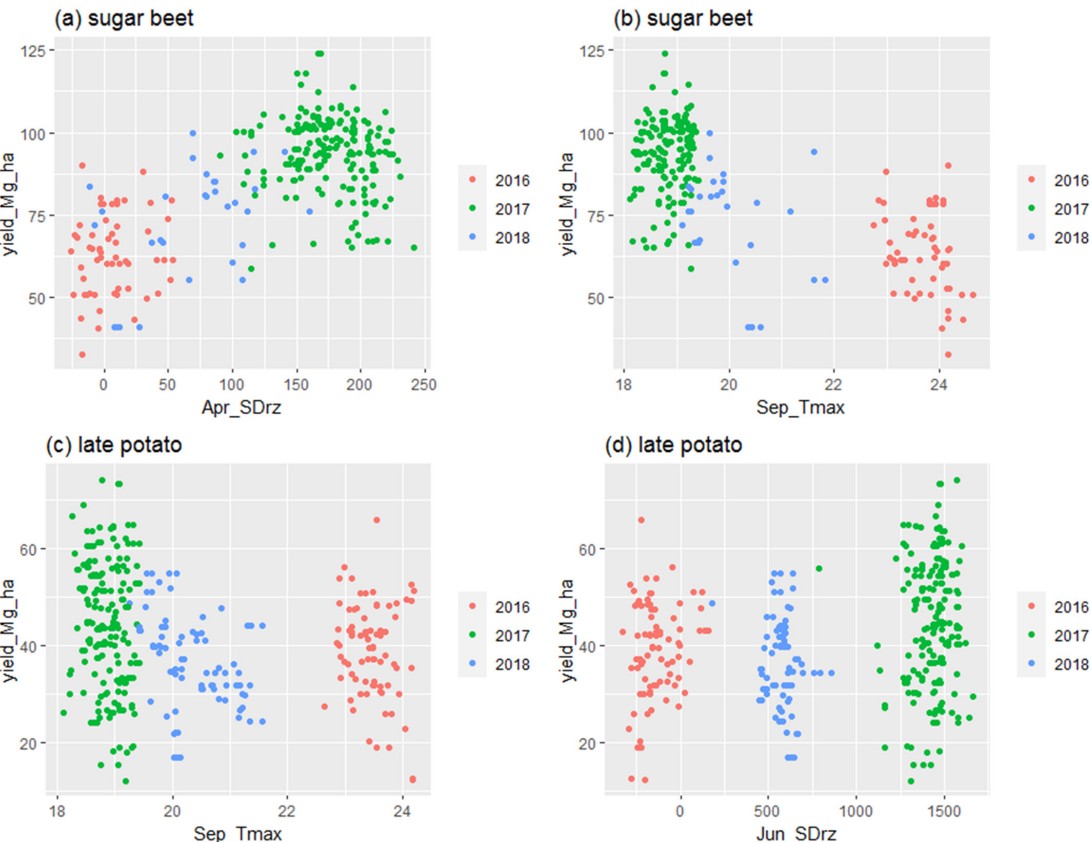

**Figure 6.** (**a**) Scatterplot: root-zone soil water depletion in April versus sugar-beet crop yield, (**b**) scatterplot: maximum temperature in September versus sugar-beet crop yield, (**c**) scatterplot: maximum temperature in September versus late-potato crop yield, and (**d**) scatterplot: root-zone soil water depletion in June versus late potato crop yield.

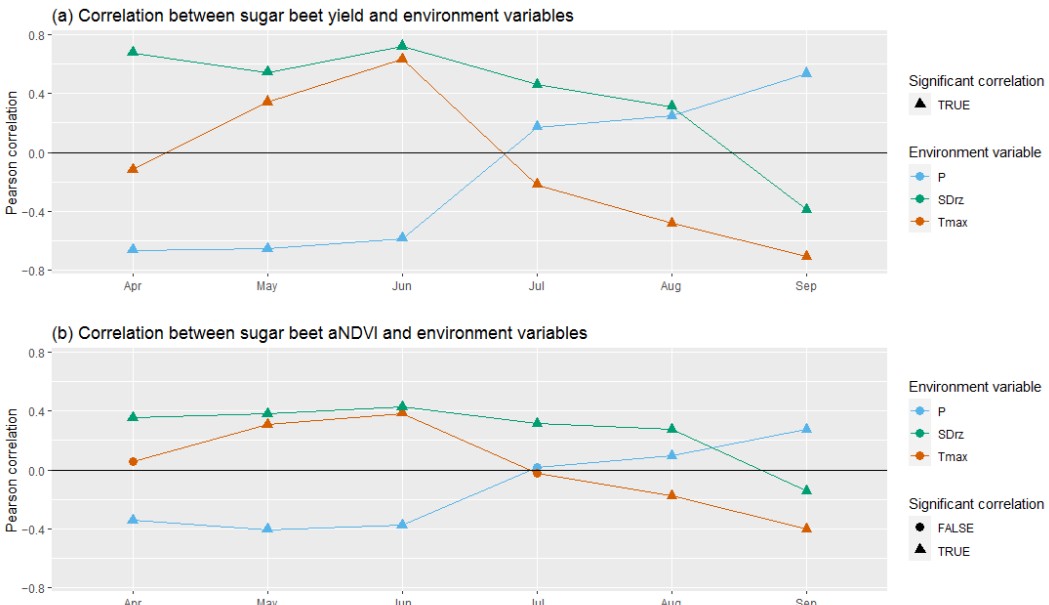

**Figure 7.** Pearson correlations for sugar beet during April–September between (**a**) yield and monthly precipitation (P), maximum temperature (Tmax), and root-zone soil water depletion (SDrz), (**b**) aNDVI and monthly precipitation (P), maximum temperature (Tmax), and root-zone soil water depletion (SDrz). Significant correlation values ($p < 0.01$) are indicated with a triangle and nonsignificant correlation values ($p > 0.01$) with a circle.

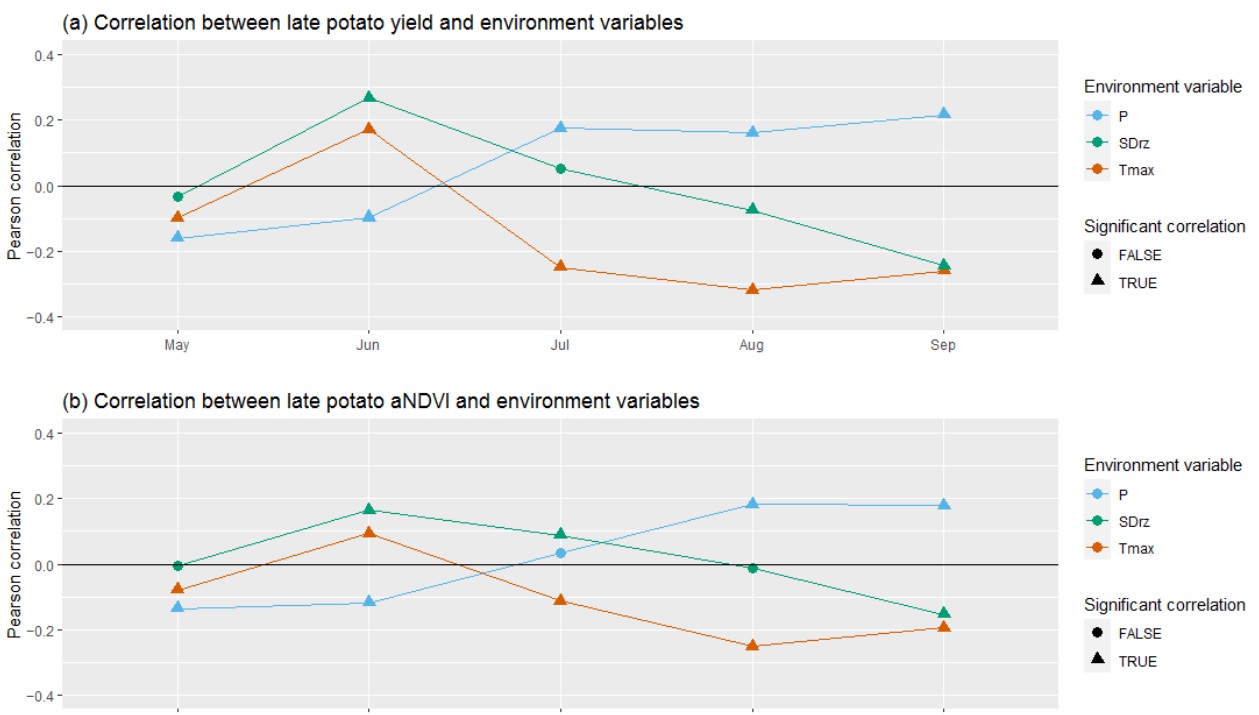

**Figure 8.** Pearson correlations for late potato during May–September between (**a**) yield and monthly precipitation (P), maximum temperature (Tmax) and root-zone soil water depletion (SDrz), (**b**) aNDVI and monthly precipitation (P), maximum temperature (Tmax) and root-zone soil water depletion (SDrz). Significant correlation values ($p < 0.01$) are indicated with a triangle and nonsignificant correlation values ($p > 0.01$) with a circle.

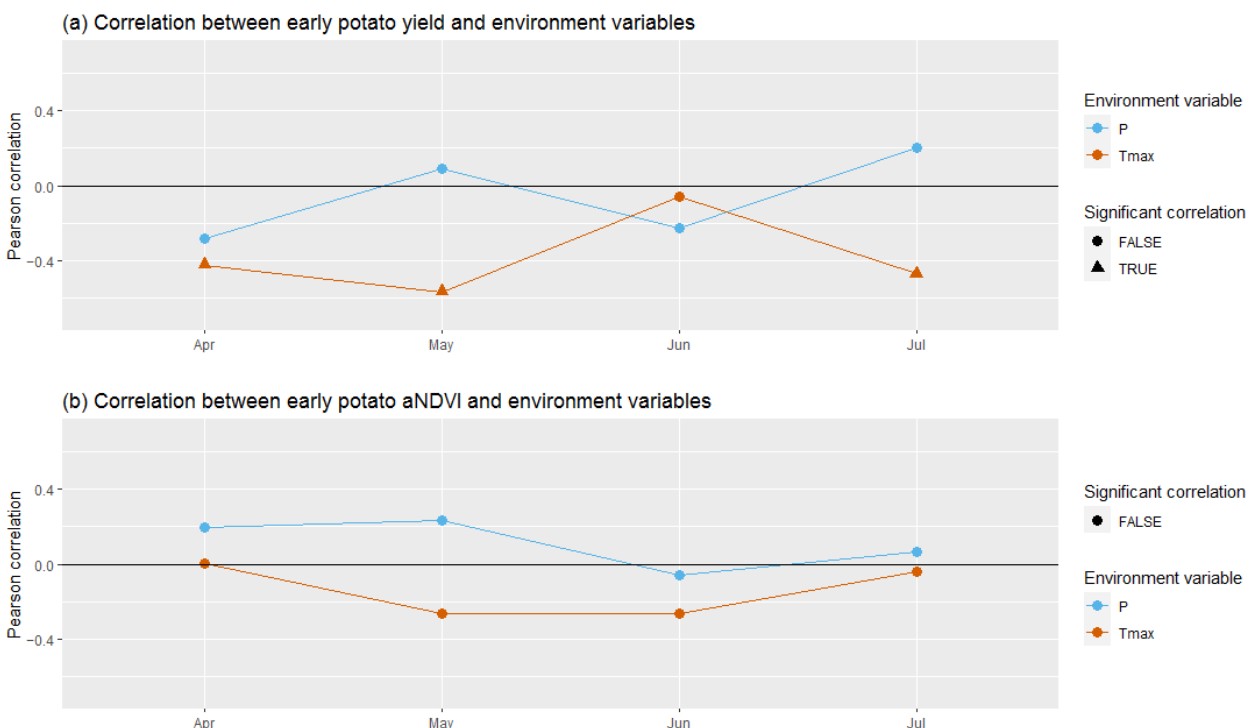

**Figure 9.** Pearson correlation for early potato during April–July between (**a**) yield, monthly precipitation (P), and maximum temperature (Tmax), (**b**) aNDVI, monthly precipitation (P), and maximum temperature (Tmax). Significant correlation values ($p < 0.01$) are indicated with a triangle and nonsignificant correlation values ($p > 0.01$) with a circle.

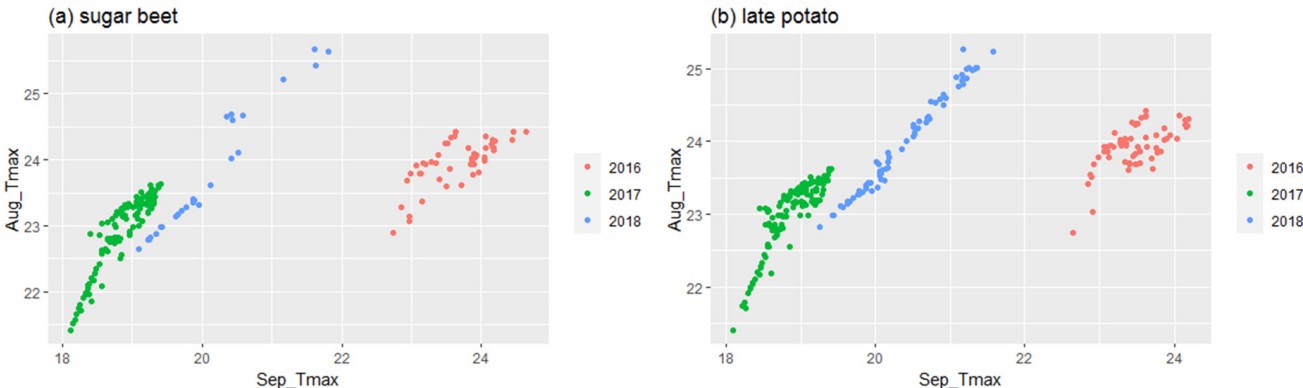

**Figure 10.** (**a**) Scatterplot: observed Tmax in August versus for Tmax in September for sugar-beet fields, (**b**) scatterplot: observed Tmax in August versus for Tmax in September for late-potato fields.

## 4. Discussion

Our results indicated that Sentinel-2-derived aNDVI was not a good predictor variable for late- and early-potato crop yields in northern Belgium (Table 2). However, in other regions potato-yield models based on NDVI achieved acceptable model performances. For example, in the Munshiganj area of Bangladesh, NDVI-based yield models were shown to explain potato-yield variability well; the yield model had a coefficient of determination ($R^2$) equal to 0.84 [26]. Salvador et al. (2020) was able to predict potato yield in Mexico based on NDVI, meteorological data (retrieved from ERA5), and the previous year's yield as predictor variables using machine-learning algorithms (random forest, support vector machine linear, support vector machine polynomial, support vector machine radial, and general linear model) resulting in models with coefficients of determination ($R^2$) reaching up to 0.86. The developed models indicated that NDVI explained a large part of the yield variability during the winter cropping season of potato in Mexico [25]. Also, in a study area in northwestern Spain, a good performing potato-yield model ($R^2 = 0.59$) based on NDVI values measured in three temporal intervals during the potato growing cycle has been developed [37]. From this we can conclude that modeling potato yields based on NDVI using an empirical model is environmentally dependent. For sugar beet, better results were obtained for the model based on aNDVI, where aNDVI explained only a small part of the sugar-beet yield variability ($R^2$ of 0.16, Table 2). However, better performing models were obtained in other regions. In a study area in Western Morocco, a yield model based on NDVI at a specific date during the growing season using a linear model reached moderate model performance; the best model reached a model performance of 0.496 [38]. Also, in Turkey, a moderate relation between NDVI at specific times during the growing season and sugar-beet yield was observed, the best linear model reached a coefficient of determination equal to 0.55 using the NDVI observed close to the end of the growing season [39]. The varying model performances of NDVI-based potato and sugar-beet yield models in different environments demonstrate that the performance of empirical yield models based on NDVI yield proxies is dependent on the environment. This confirms earlier findings for winter wheat [8,9]. Therefore, it is important to evaluate whether NDVI series are sensitive to yield-affecting environment conditions in order to be able to use the indicator as a predictor in potato and sugar-beet empirical yield models in a specific environment.

The low model performance of potato and sugar-beet yield models based on aNDVI only might also be related to the high yields of sugar beet and potato in Belgium. The average yield was 41 Mg/ha for potato and 88 Mg/ha for sugar beet in Belgium, whereas the average yield in Europe was 23 Mg/ha for potato and 62 Mg/ha for sugar beet in 2019 [40]. The authors of [16] found that for soybean and corn yield, the performance of models based on vegetation indices decreased with increasing yield. This was related to saturation of multispectral data (including NDVI) at high yields [16]. For late potato, our

model also suggests this, since yields were underestimated at higher yields, whereas late potato yield was overestimated at lower yields (Figure 5b).

Adding monthly weather variables to the crop-yield models improved the yield model performance for all three crops remarkably, indicating that weather variables explained a large part of the crop-yield variability of sugar beet, late potato, and early potato in northern Belgium. When soil texture information at the field level was considered, by means of modeling crop yield in function of aNDVI and monthly root-zone soil water depletion throughout the growing season, the performances of the sugar-beet and late-potato models were similar to when aNDVI and weather information were included in the crop-yield models (Models 2 and 3, Table 2). When both weather variables and root-zone soil water depletion throughout the growing season in combination with aNDVI were used as predictor variables the model performance was not higher compared to when only weather variables or root-zone soil water depletion in combination with aNDVI were used to model sugar-beet and late-potato crop yield. However, Tmax and SDrz during certain months in combination with aNDVI explained a large part of the sugar-beet ($R^2$ = 0.84) and late-potato ($R^2$ = 0.56) yield variability (Figures 4 and 7). The findings on the contribution of environmental variables to sugar-beet and potato yield variability were confirmed by earlier research. A combined soil–water balance and biomass model, based on biometeorological data, captured up to 84% of the variation in observed sugar-beet yields and 83% of late-potato yield variation during 1960–2008 [41] and elucidated the adverse impact of moisture and temperature related stress [21]. More recently, the impact of extreme meteorological events during the sensitive stages of potato and sugar beet pointed to the importance of temperature and rainfall related variables [42]. The spatio-temporal variability of (extreme) dry and wet spells elucidated significant ($p < 0.001$) effects of wet spells on sugar-beet yields, whereas (extreme) dry spells significantly ($p < 0.001$) affected potato yields [43].

The novelty of this research lies in the combination of low temporal resolution meteorological variables and the remote-sensing-derived indicator NDVI, both of which are commonly available and commensurate with the regional-scale assessment of crop performance and yield. The correlation patterns between crop yield and monthly environmental variables showed a similar pattern as the correlation between aNDVI and monthly environmental variables for sugar beet, late potato, and early potato (Figures 7–9). This may indicate that aNDVI is affected by monthly P, Tmax, and SDrz in a similar way as crop yield. However, for late and early potato, the correlation values between aNDVI and the monthly environmental variables were lower compared to the correlation values between yield and the monthly environmental variables. This suggests that the calculated aNDVI for late- and early-potato fields was not strongly affected by the same environmental variables that affect potato yield. For early potato, none of the correlation values between aNDVI and the monthly environmental variables were significant (Figure 9). This could explain why the crop model based on aNDVI only (Model 1) reached acceptable model performances for sugar beet but not for late and early potato.

For sugar beet the root-zone soil water depletion in April explained a large part of the crop yield variability (Figure 4). According to the Pearson correlation plot, higher SDrz values for the month of April were correlated with higher sugar-beet yields (Figure 7). Modeled SDrz for April were close to zero and sometimes negative for some fields in 2016 and 2018 (Figure 6), indicating that soils were close to saturation and waterlogging may have occurred on these fields resulting in lower sugar-beet establishment and therefore yields. Low sugar-beet yields caused by waterlogging during the seeding and germination was also observed by [42]. For late-potato fields, SDrz in June was close to zero and sometimes negative in 2016 (Figure 6) indicating that soils were close to saturation and waterlogging may have occurred on these fields. Potato yield is known to be sensitive to waterlogging due to the shallow rooting system of potato plants [42,44]. The negative Pearson correlation between SDrz in June and potato yield reflects this. SDrz in June explained a large part of the late potato yield variability.

Maximum temperatures in September were an important predictor variable in both the sugar-beet and late-potato crop-yield models (Figure 4). Both sugar-beet and late-potato yield were negatively correlated with maximum temperatures in September (Figures 7 and 8). The importance of the variable Tmax in September in the crop-yield models probably reflects the negative effect of cumulative drought stress in August and September on crop yield rather than the effect of Tmax in September only. Fields which experienced a high Tmax in September also experienced a high Tmax in August, which in turn was negatively correlated with sugar-beet and late-potato yield (Figures 7, 8 and 10). Elevated temperatures late in the growing season are known to affect sugar-beet yield negatively [42,45–47]. A reduction of 11% of total dry sugar-beet biomass due to late season elevated temperatures was related to decreased light interception owing to an accelerated decline in leaf area index, whereby an increased maintenance respiration was reported [46]. Potato tuber yield and quality were negatively affected by elevated temperatures [42,48,49]. Hollow hearts, cracking, and secondary growths of potato are examples of temperature-induced tuber malformations caused by heat stress late in the growing season [49].

## 5. Conclusions

In this research we demonstrated that models based on the NDVI integral only were not able to explain early- and late-potato yield variability and only partly explained sugar-beet yield variability from 2016–2018 in northern Belgium. The NDVI series of early and late potato were not sensitive enough to yield affecting weather and soil water conditions during particular phenological stages. Random forest regression based on commonly available NDVI, monthly temperature, and monthly rainfall explained up to 57% of late potato, 68% of early-potato, and 85% of sugar-beet yields. We found that water-saturated conditions early in the growing season, i.e., April for sugar beet and June for late potato, and elevated temperatures late in the growing season, i.e., September, explained a large part of the sugar-beet and late-potato yield variability. Our findings confirmed the importance of meteorological and soil water condition variables during sensitive phenological stages. We concluded that yield affecting weather and soil water conditions during sensitive phenological stages are needed in addition to the NDVI integral to be able to model the crop-yield variability of sugar beet and potato in northern Belgium using empirical crop models.

**Author Contributions:** Conceptualization and methodology, A.G. and A.V.; validation, formal analysis, writing—original draft preparation, A.V.; data curation and writing—review and editing, A.G. and A.V.; supervision, A.G., funding acquisition, A.G. All authors have read and agreed to the published version of the manuscript.

**Funding:** The authors acknowledge funding from the European Union's Horizon 2020 Research and Innovation Programme under grant agreement No. 818346.

**Data Availability Statement:** NDVI Sentinel-2 timeseries can be downloaded using the https://openeo.org/platform, accessed on 1 July 2021. Crop-yield and agricultural parcel information were made available through the Department of Agriculture and Fisheries. Weather data were made available through the Royal Meteorological Institute.

**Acknowledgments:** The authors are grateful to the Royal Meteorological Institute for weather data, and to the Department of Agriculture and Fisheries for agricultural parcel and yield data.

**Conflicts of Interest:** The authors declare no conflict of interest.

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
