# Peer review of "Estimating Yield from NDVI, Weather Data, and Soil Water Depletion for Sugar Beet and Potato in Northern Belgium"

_water, doi:10.3390/w14081188_

Round 1
Reviewer 1 Report
The manuscript is good. Though cases of plagiarism were detected. The manuscript should be re-written to that it is free of plagiarism.

Author Response
Dear reviewer,
We appreciate the interesting questions and remarks on our manuscript. We revised the manuscript with care and added the reference to the two manuscripts which were indeed lacking from our reference list: see lines 108-134 and 47.
All the best,
The authors
Reviewer 2 Report
I believe that the manuscript is worth publishing in this journal with revision.
This study evaluated the model performance of NDVI based on random forest models for sugar beet and potato farm yields in northern Belgium. The authors draw the conclusion of that NDVI integral in combination with monthly precipitation, maximum temperature and soil water depletion during the growing season explained sugar beet and late potato yield variability well. It will be important for model use in the future in this region.
- Line 10: Define abbreviation NDVI when it is first used in the manuscript.
- The performance of empirical potato and sugar beet crop yield models based on the remotely sensed vegetation index NDVI has been evaluated in northern Belgium, in this text. How about the rest of the world areas. Is there and suggestions to other countries?
- Discussion should be based on more literatures to compare your results with results acquired.
- NDVI is susceptible to atmospheric noise and is prone to saturation under high LAI values. In view of this problem, whether the model is corrected and evaluated to avoid it should be considered.
- Field irrigation is a key measure in determining crop yields. What is the basis for neglecting it? (line 127-128: Irrigation was assumed zero for all fields.).
- I can't find Table 3 anywhere in the text. (lines 275 & 290)
- The conclusion is too long. State and summarize the main points.
Author Response
Dear reviewer,
We appreciate the interesting questions and remarks on our manuscript. We revised the manuscript with care and provide a response to your remarks below.
All the best,
The authors
- Line 10: Define abbreviation NDVI when it is first used in the manuscript.
This was added
- The performance of empirical potato and sugar beet crop yield models based on the remotely sensed vegetation index NDVI has been evaluated in northern Belgium, in this text. How about the rest of the world areas. Is there and suggestions to other countries?
In previous research we found that model evaluation metrics of winter wheat yield models based on NDVI yield proxies is dependent on the location. (see lines 65-70). When we compare our results to previous research (see lines 276-305) we found that this was also the case for potato and sugar beet. We added some sentences on this on lines 276-305.
- Discussion should be based on more literatures to compare your results with results acquired.
We added some references. See lines 276-314
- NDVI is susceptible to atmospheric noise and is prone to saturation under high LAI values. In view of this problem, whether the model is corrected and evaluated to avoid it should be considered.
This is indeed a problem of NDVI. We did not correct for this. The NDVI saturation might also be an explanation of the low model performance of models based on aNDVI only. We added some sentences on this on lines 307-314.
- Field irrigation is a key measure in determining crop yields. What is the basis for neglecting it? (line 127-128: Irrigation was assumed zero for all fields.).
Irrigation is not common in northern Belgium for the crops sugar beet and potato. Only under extremely dry circumstances farmers will apply irrigation (i.e. supplementary irrigation). We added some sentences on this in lines 127-129.
- I can't find Table 3 anywhere in the text. (lines 275 & 290)
Reference was changed, sentences refer to Table 2.
- The conclusion is too long. State and summarize the main points.
We rewrote the conclusion so that it only includes the main points.
Round 2
Reviewer 2 Report
It can be accepted now.